# Tuberculosis Infection in Pregnant People: Current Practices and Research Priorities

**DOI:** 10.3390/pathogens11121481

**Published:** 2022-12-06

**Authors:** Jyoti S. Mathad, Sharan Yadav, Arthi Vaidyanathan, Amita Gupta, Sylvia M. LaCourse

**Affiliations:** 1Department of Medicine, Center for Global Health, Weill Cornell Medicine/New York Presbyterian Hospital, New York, NY 10065, USA; 2Department of Obstetrics and Gynecology, Weill Cornell Medicine, New York, NY 10065, USA; 3Department of Medicine, Division of Infectious Diseases, Johns Hopkins School of Medicine, Baltimore, MD 21205, USA; 4Departments of Medicine, Global Health, and Epidemiology, Division of Allergy and Infectious Diseases, University of Washington, Seattle, WA 98104, USA

**Keywords:** tuberculosis, prevention, pregnant, HIV, isoniazid, postpartum

## Abstract

Women are significantly more likely to develop tuberculosis (TB) disease within the first 90 days after pregnancy than any other time in their lives. Whether pregnancy increases risk of progression from TB infection (TBI) to TB disease is unknown and is an active area of investigation. In this review, we discuss the epidemiology of TB and TBI in pregnancy, TBI diagnostics, and prevalence in pregnancy. We also review TBI treatment and highlight research priorities, such as short-course TB prevention regimens, drug-resistant TB prevention, and additional considerations for safety, tolerability, and pharmacokinetics that are unique to pregnant and postpartum people.

## 1. Introduction

Of the 10 million new tuberculosis (TB) diagnoses each year, one third are in women [1]. Most TB diagnoses in women occur between the ages of 15 and 49 years, which overlaps with years typically associated with reproductive potential [2]. However, epidemiologic data about TB disease during pregnancy are not routinely collected. In the absence of systematically collected data, global modeling studies estimate 200,000 incident TB diagnoses during pregnancy each year [3].

Whether pregnancy increases risk of progression from TB infection (TBI) (a state of persistent immune response to stimulation by *Mycobacterium tuberculosis* (Mtb) antigens without evidence of clinically active TB) [4] to TB disease is unknown [5]. Large epidemiologic studies suggest women are significantly more likely to develop TB disease within the first 90 days after pregnancy than any other time in their lives [6,7]. A United Kingdom study followed 192,000 people who became pregnant with a TB incidence rate ratio of 1.95 (95% CI 1.24–3.07) postpartum compared to nonpregnant times [6]. Similarly, a large study in Sweden reviewed 649,000 medical records of women in their reproductive years and found increased incidence of TB during pregnancy (IRR 1.4, 95% CI 1.1–1.7) and within 6 months postpartum (IRR 1.9, 95% CI 1.5–2.5) compared to nonpregnant periods [7]. In studies of women with HIV prior to and since widespread antiretroviral therapy (ART), even higher postpartum TB incidence and mortality are reported [8,9]. 

It is critical to prevent TB disease during pregnancy. Maternal TB disease is associated with poor outcomes including miscarriage, pre-eclampsia/eclampsia, low birthweight, premature birth, and mortality, in addition to TB transmission to the infant and family [10]. These outcomes are exacerbated by HIV, even in the setting of widespread ART [11], and lack of access or delay in accessing health care. Malnutrition, alcohol or tobacco abuse, and diabetes may also contribute to poor TB outcomes during pregnancy, as they do in non-pregnant populations, but there are limited data on these risk factors in pregnant people.

## 2. Epidemiology of TB Infection in Pregnancy

One fourth of the global population is estimated to have TBI [12]. Inconsistent screening and lack of an established gold-standard of TBI testing in pregnancy prevent accurate estimates [10]. Table 1 summarizes studies of TBI prevalence in pregnant people from high- and low-burden settings. 

### 2.1. High-Burden Settings

In high-burden settings, most studies report TBI prevalence in pregnancy between 30 and 34%, including for people with HIV [8,9,13,14,15,16,17,18,19,20,21,22,23,24]. TBI detected through tuberculin skin testing (TST) ranges from 4.6 to 35.6% and from 18.9 to 37.8% [8,9,13,14,15,16,17,18,19,20,21,22,23,24] using interferon gamma release assays (IGRA) (Table 1). 

### 2.2. Low-Burden Settings

In low-burden TB settings, estimated TBI prevalence is expectedly lower, with a median of 15% by TST (range 4.2–50%) and 11.4% by IGRA (range 2–22%) [25,26,27,28,29,30,31,32,33,34,35,36,37]. Studies reporting TBI prevalence >25% in pregnancy in low-burden settings are usually conducted in higher-risk populations, such as pregnant people from high-burden TB settings (Table 1) [10,29]. 

**Table 1 pathogens-11-01481-t001:** TBI prevalence and diagnostics.

**High-Burden Regions**								
**Authors and Date**	**Study Location**	**Purpose**	**Research** **Design**	**N**	**% HIV, Median CD4**	**Prevalence**		**Findings**
						**TST, %**	**IGRA, %**	
Weinberg et al., 2021 [13]	Botswana, Haiti, India, South Africa, Tanzania, Thailand, Uganda, Zimbabwe	Determine the effect of pregnancy stage and IPT on IGRA and TST	Randomized, double blinded, placebo controlled trial	944	100; Median CD4: 521		30 ^a^	From pregnancy to delivery, 24% of IGRA-positive women reverted to IGRA-negative or indeterminate; 62% became IGRA positive again postpartum. Loss of IGRA positivity during pregnancy explained by decreased IFN-γ production and IPT. TST less affected by pregnancy but had lower positivity compared to IGRA at all time points.
König Walles et al., 2018 [14]	Ethiopia	Determine diagnostic yield of TB2 and the agreement between TB1 and TB2 results in pregnant people tested with QFT-Plus.	Prospective cohort	829	5.9; Median CD4 range: 400–799		33 ^a^	High agreement between QFT-Plus results elicited by TB1 and TB2 antigen formulations, including in WHIV. IFN-γ responses among HIV-positive women were significantly lower than those in HIV-negative persons, suggesting that a lower cutoff might be considered to define positive QFT-Plus results for HIV-positive pregnant women.
Birku et al., 2020 [15]	Ethiopia	Effect of pregnancy and HIV infection on performance of TST and IGRA	Cross-sectional	159	54; Median CD4: 384	HIV−: 35.6, HIV+: 20.9	HIV−: 31.5 ^a^, HIV+: 18.9 ^a^	HIV infection reduces rate of detection of TBI by TST and IGRA during pregnancy. Concordance between TST and IGRA increased w/pregnancy and/or HIV infection; overall agreement 92.4%, κ = 0.82.
Walles et al., 2021 [16]	Ethiopia	Study TB exposure patterns in women of reproductive age	Prospective cohort	2088	9.3; Median CD4 range: 350–599		33 ^a^	TBI associated with age and HIV infection. Without HIV, absolute annual risk of acquiring TB infection was 2.1%.
Tesfaye et al., 2021 [17]	Ethiopia	Compare Mtb-triggered IFN-γ levels longitudinally in pregnant and postpartum women	Prospective cohort	363	0		22 ^a^	Mtb-stimulated IFN-γ responses were higher during 3rd trimester than earlier stages of pregnancy and postpartum, despite decreased mitogen-triggered responses.
Gupta et al., 2007 [9]	India	Determine the incidence of active TB in postpartum WHIV	Prospective cohort	688	100; Median CD4: 465	21		TB incidence was 5 cases per 100 person-years. Predictors of incident TB included CD4 count <200, HIV viral load <50,000 copies/mL, and positive TST. Women with incident TB and their infants had a 2.2- and 3.4-fold increased probability of death, respectively.
Mathad et al., 2014 [18]	India	Examine how pregnancy impacts the performance of TBI diagnostics	Cross-sectional + longitudinal cohort	401	0	14	37 ^a^	Agreement 76%, κ = 0.37. Pregnancy stage affected both IGRA and TST, but IGRA was more sensitive and stable. Median IFN-γ concentration lowest at delivery and highest postpartum.
Mathad et al., 2016 [19]	India	Performance of TBI tests in pregnant and postpartum WHIV, investigate the immunology behind discordance	Cross-sectional + longitudinal cohort	252	100; Median CD4: 468	10	28 ^a^	Agreement 75%, κ = 0.25 QGIT was more stable and likely more accurate than TST. Pregnant women with IGRA+/TST− discordance had less IFN-γ and IL-2 than those with concordant-positive results and may represent a high-risk subset for the development of active TB postpartum.
Bhosale et al., 2021 [20]	India	Compare performance of TBI tests over time in pregnant people with and without HIV	Prospective cohort study of IGRA+ pregnant people	165	21; Median CD4: 476			Pregnancy affects TBI test results and reduces IFN-γ response to Mtb stimulation. IGRA/TST discordance high in pregnant women (HIV+: 51%; HIV−: 25%). Despite adequate CD4 counts, WHIV express less IFN-γ than women without HIV.
Jonnalagadda et al., 2010 [8]	Kenya	Estimate sensitivity, specificity, and positive predictive value of IFN-γ and CD4 for postpartum TB	Cohort	333	100; Median CD4: 440		36 ^b^	Positive IGRA results for pregnant WHIV were associated with postpartum active TB (aHR_CD4_, 4.5; 95% CI 1.1–18.0).
LaCourse et al., 2017 [21]	Kenya	Determine the effect of peripartum stage on TST and IGRA	Prospective cohort	100	100; Median CD4: 555	13.5	35.4 ^b^	Low agreement between QFT and TST (κ = 0.20); QFT identified >2-fold more women with TBI compared to TST in pregnancy and postpartum. Lower QFT Mtb-Ag and mitogen responses in pregnancy compared to postpartum.
Kaplan et al., 2022 [22]	Kenya	Estimate TBI prevalence and effect of HIV on diagnostic performance of IGRA versus TST	Cross-sectional	400	50; Median CD4: 464	HIV–: 4.6 HIV+: 18.5	HIV–:33.2 ^a^, HIV+: 31.5 ^a^	QFT-Plus identified 3-fold more women with TBI compared with TST. Difference further amplified when increasing TST cutoffs to ≥10 mm for both WLHIV- and HIV-negative women. QFT-Plus positivity prevalence was similar regardless of HIV status, although TB-specific antigen responses lower with HIV.
Sheriff et al., 2010 [23]	Tanzania	Prevalence of TBI in pregnancy	Cross-sectional	286	4.5	14.5		TBI prevalence 26.2–37.4%. Certain ethnic groups were less vulnerable to TBI compared to others. Age, parity, HIV status, BMI did not affect TST results.
Bongomin et al., 2021 [24]	Uganda	TBI prevalence and risk factors among pregnant women	Cross-sectional	261	5		37.8 ^a^	HIV infection and ages 30–39 were independently associated with TBI (OR 4.4, *p* = 0.04, and OR 4.0, *p* = 0.02, respectively).
**Low-Burden Regions**								
**Authors** **and Date**	**Study Location**	**Purpose**	**Research** **Design**	**N**	**HIV Status, Mean CD4**	**Prevalence**		**Findings**
						**TST, %**	**IGRA, %**	
Fröberg et al., 2020 [25]	Sweden	To evaluate the newly introduced TB screening program among pregnant women in Stockholm in 2016–2017.	Retrospective, observational study	~2000	0.2		22 ^a^	TBI treatment among all QFT-positive pregnant women increased from 24% to 37%. Treatment completion (mainly rifampicin) postpartum was 94%. No WHIV were treated for TBI. Nine HIV-negative active pulmonary TB cases were detected (incidence: 215/100,000). None had been screened for TB prior to pregnancy.
Present, Comstock 1975 [26]	USA	To determine if pregnancy affects TST sensitivity	Case-control study	452	Not Reported			No indication that pregnancy affects the level of TST sensitivity.
Covelli, Wilson 1987 [27]	USA	To assess changes in cell-mediated immunity in pregnancy	Cohort study	172	Not Reported	12.2		Progressive depression of lymphocyte response to TST, from 36 weeks’ gestation through delivery. Nonspecific cell-mediated immunity was maintained.
Mofenson et al., 1995 [28]	USA	Prevalence of TBI and anergy among pregnant and non-pregnant WHIV	Prospective study	46	100; Mean CD4: 530	11		Prevalence of anergy was higher in nonpregnant vs. pregnant women (42% vs. 38%).
Medchill 1999 [29]	USA	To evaluate a TST program in an obstetric clinic	Retrospective chart review	1497	0	15		TST positivity was 15.2%: 1.3% of Asian, 23.9% of Hispanic, 9.3% of black, and 4.1% of white patients. Hispanic patients had a relative risk for positive TST of 5.9 compared with white patients.
Jackson 2001 [30]	USA	Prevalence of anergy in pregnant versus nonpregnant women	Case-control	60	0	10		Pregnant women were less likely to have a reaction to tetanus toxoid than nonpregnant women (10% vs. 40%; *p* < 0.02). Otherwise, no difference in anergy between pregnant and non-pregnant.
Schulte et al., 2002 [31]	USA	To determine the number of pregnant WHIV that completed TST	Retrospective chart review	176	100	26		A total of 85% of women completed TST; 21% had positive TSTs, and 1% had active TB disease
Cruz et al., 2005 [32]	USA	To investigate rates and predictors for follow up and treatment among postpartum women with positive TST	Retrospective cohort	1131	0	32		Only 18% of patients with positive PPD completed 6 months of therapy. Those who received care from the same physician antepartum and postpartum were more likely to attend and complete therapy (67% (*p* < 0.01) and 62% (*p* = 0.1), respectively)
Schwartz et al., 2009 [33]	USA	To review compliance and utility of universal TB screening among pregnant women	Retrospective data review	3847	0	50		A total of 95% of patients were compliant with TST testing; 50.4% had positive PPD results, of which 95.1% completed chest X-rays.
Chehab et al., 2010 [34]	USA	To assess the consistency of TST and QFT testing among pregnant and non- pregnant women	Case-control	152	0	10	7 ^a^	Overall concordant results: agreement 86.2% (κ = 0.288). More discordant results in non-pregnant women (24% vs. 8.8%; *p* < 0.022); most commonly TST+/QFT-ve.
Worjoloh et al., 2011 [35]	USA	To estimate agreement between TST and IGRA in pregnant women.	Cross-sectional	199	0	23	14 ^a^	Poor concordance and agreement between TST and IGRA (77.3%, κ = 0.26). Most common discordance: TST+/QFT−. No association between gestational age and IGRA positivity or discordant results. Changing IGRA cutoff did not improve agreement.
Lighter-Fisher & Surette et al., 2012 [36]	USA	To evaluate IGRA for TBI in pregnant adolescents and high-risk women	Prospective	280	Not Reported	21	11.4 ^a^	IGRA associated with greater likelihood of exposure to Mtb. IFN-γ did not have any temporal association with pregnancy stage. Agreement between QFT and TST in pregnant women = 88%, κ = 0.452.
Molina et al., 2016 [37]	USA	Completion rates and concordance between IGRA and TST in pregnancy	Observational cohort	141	2.8	4.2	2 ^b^	IGRA had a higher completion rate than TST (98% vs. 63%) while maintaining high concordance (96.3%).

^a^ QuantiFERON-TB Gold or QFT-Plus, ^b^ T-Spot.TB; Abbreviations: IPT = Isoniazid preventative therapy; IRGA = Interferon gamma release assays; TST= Tuberculin skin test; QFT = QuantiFERON; WHIV = Women with HIV; QGIT = QuantiFERON-Gold-in-tube; PPD = Purified protein derivative (form of TST).

## 3. Natural History of TB in Pregnancy

Few studies specifically assess how the immunologic changes of pregnancy affect TB risk. During pregnancy, cells associated with cell-mediated immunity (e.g., CD4+/CD8+-T cells) decrease while other cells that dampen the immune response (e.g., Treg) increase in frequency [38]. Some data suggest CD4+ and CD8+ T-cell function is also decreased during pregnancy [39]. The relative immune suppression associated with pregnancy progresses until a nadir at delivery [38]. These subtle changes increase the risk of certain infections (e.g., Listeria) and severity of other infections (e.g., influenza). There are no definitive studies on how these changes specifically affect susceptibility to Mtb.

Studies from both Kenya and India document a decrease in quantitative interferon gamma (IFN-γ) produced after ex vivo stimulation with Mtb-specific antigens of samples during pregnancy versus postpartum [18,19,20,21]. Similar patterns are seen in pregnant people with and without HIV, although people with HIV have lower IFN-γ production at all stages of pregnancy, despite adequate CD4+ T-cell counts [20,22]. A Mtb-specific CD4+ polyfunctional response was also decreased in late pregnancy among women in Kenya and Uganda with and without HIV [40]. Interestingly, a study from India reported that pregnant people with gestational diabetes (GDM) had an impaired IFN-γ response to Mtb-specific antigens; the impairment was highest in pregnant people with GDM and HIV [41]. These data suggest that impairment of cell-mediated immunity during pregnancy may be exacerbated by comorbidities such as HIV and GDM and may allow TBI to progress to disease. Further research on the intersection of GDM, HIV, and TB in pregnant people are warranted.

## 4. TB Infection Diagnostics

TBI screening is recommended for pregnant people with HIV, other immunosuppression (e.g., chronic steroid use, TNF-alpha inhibitors, cancer, and/or chemotherapy), or recent contact with someone with pulmonary TB disease [42,43]. Currently available tests to detect TBI include IGRA and TST; both are safe in pregnancy. 

Both IGRA and TST rely on a functioning cell-mediated immune system, which is affected by pregnancy. Currently, there are no guidelines suggesting modification of TST cutoffs based on pregnancy alone. Two studies conducted in the United States noted no significant effect of pregnancy status or stage on TST results, with similar rates of cutaneous anergy among pregnant and non-pregnant people with HIV [30]. However, studies in TB-endemic regions showed a decreased TST response during pregnancy versus postpartum [13,18,21,22].

Numerous studies have been performed comparing performance of IGRA and TST in pregnant and postpartum people. In low-TB-prevalence regions, most studies report moderate concordance [20,44,45,46]. IGRA has increased completion rates [16] and may have higher specificity [13,16,36], particularly in people with prior BCG vaccination [25], compared to TST. In high-TB-burden regions, data on concordance between IGRA and TST are mixed. Most studies report mild to moderate agreement and concordance [14,15,17,20,21]. The most common type of discordance is IGRA+/TST− [14,15,17,20,21]. IGRA+/TST− discordance may occur because TST requires multiple cytokines to increase to trigger induration versus the IGRA, which is based only on IFN-γ levels. Test performances also fluctuate with gestational age [13,17,18,19,21] and HIV status [20,22]. One study found that pregnant women with HIV and IGRA+/TST− discordance had lower IFN-γ and IL-2 in response to Mtb-specific antigens compared to IGRA+/TST+ results. Furthermore, the majority of women who developed TB disease postpartum had IGRA+/TST− results during pregnancy, suggesting that discordance, itself, may predict disease [19]. Larger studies are needed. 

The nadir of positivity for both tests is at delivery, coinciding with the nadir of measured IFN-γ levels, with highest positivity noted in the postpartum period [13,17,18,19,20,21]. This may explain the high incidence of IGRA reversion from positive to negative or indeterminate at delivery, with conversion back to positive tests postpartum [13,17,18,19,20,21].

### Pregnant People with HIV

IGRA positivity has been shown to be 2–3 times higher than TST positivity at every stage of pregnancy including postpartum among women with and without HIV [18,19,21,22]. Compared to pregnant people without HIV, IFN-γ levels are lower among pregnant people with HIV, despite being on ARTs with adequate CD4+ T-cell counts [20,22]. A functional impairment in the ability of Mtb-specific CD4 T-cells to produce IFN-γ, even with chronic ART [47], may explain lower TST and IGRA positivity in these populations [20,22].

Larger studies are needed to determine the optimal type and timing of TBI testing during pregnancy. Until then, all pregnant people with a positive IGRA or TST should be assessed for TB disease, which can include a symptom screen and sputum sample for molecular testing and/or a Mtb culture. A chest radiograph with an abdominal shield is safe for the fetus and should be performed whenever thoracic TB disease is suspected [48]. If there is no evidence of TB disease, then women with TBI should be managed as discussed below.

## 5. TB Infection Treatment 

Until recently, there were no systematic evaluations of TB preventive therapy in pregnant people. While efficacy is presumed to be similar, there are additional considerations for safety, tolerability, and pharmacokinetics in pregnant compared to nonpregnant people.

Recommended TBI regimens in pregnancy are reviewed in Table 2. Both the World Health Organization (WHO) and the United States Centers for Disease Control and Prevention (CDC) recommend isoniazid (6 or 9 months (6H or 9H)) as the preferred TB preventive therapy for pregnant people with HIV [4,43]. The WHO further recommends that 36 months of isoniazid be used for people in high-burden areas. The US CDC also includes rifampin daily for 4 months (4R) or 3 months daily isoniazid and rifampin (3HR) as alternative TBI regimens during pregnancy [49,50]. Some experts prefer 4R to avoid hepatotoxicity associated with isoniazid, while others prefer isoniazid to avoid drug-drug interactions with rifampin, such as with some antiretroviral medications. 

TBI treatment recommendations during pregnancy are primarily based on safety and efficacy data on nonpregnant populations [51]. Of the currently recommended regimens, only 6H has been evaluated specifically in pregnant people in a randomized control trial. TB APPRISE evaluated the safety of 6H given during pregnancy (immediate arm) or 12 weeks after delivery (deferred arm) [52]. While maternal adverse events (or permanent discontinuation due to toxic effects) (15.03 vs. 14.93 events/100 person-years) and TB incidence (0.60 vs. 0.59/100 person-years) were similar between arms, the immediate arm had a higher risk of composite adverse pregnancy outcome (stillbirth or spontaneous abortion, low birth weight, preterm delivery, or congenital anomalies) compared to the deferred arm (23.6% vs. 17.0%, respectively; 6.7% difference (95% CI 0.8–11.9%). 

In contrast, two observational studies of isoniazid given to pregnant women with HIV in programmatic settings in South Africa did not find an association of isoniazid with adverse pregnancy outcomes [11,53]. Similarly, participants who became pregnant in the BOTUSA trial (36 months of isoniazid for people with HIV) in Botswana did not have increased adverse pregnancy outcomes [54]. A subsequent systematic review confirmed the inconsistent associations between isoniazid and adverse pregnancy outcomes, including hepatotoxicity, among pregnant people with HIV [55]. A meta-analysis focusing on individual data from clinical trials of pregnant participants with HIV and receiving TB preventive therapy is ongoing.

### Breastfeeding Considerations 

The low concentrations of isoniazid and rifampin in breastmilk are considered safe for infants [56] but not therapeutic. If the infant is in contact with a person with pulmonary TB disease, the infant should be dosed for TB prevention. Rifampin (and rifamycins in general) can cause red-orange coloration of body fluids, including breastmilk, which is not harmful to the lactating parent or infant.

## 6. Newer Short-Course TB Treatment Regimens in Pregnant People

Newer short-course rifapentine-based regimens, including isoniazid and rifapentine weekly for 3 months (3HP) and daily for 1 month (1HP), are not currently recommended for people who are pregnant or anticipating to become pregnant during TBI treatment due to lack of safety data [1,2,3]. IMPAACT 2001, a phase I/II trial, evaluated the pharmacokinetics and safety of 3HP during pregnancy [57]. Of 50 participants, 20 had HIV and were taking efavirenz-based ART. Despite lower rifapentine clearance during pregnancy vs. postpartum in women without HIV, and higher clearance in women with HIV vs. without HIV during pregnancy, all women met target exposures of rifapentine and isoniazid associated with successful TB prevention in non-pregnant cohorts. This study was not powered for safety, but there were no major drug-related safety issues identified. Similarly, data from 87 participants who became pregnant during the PREVENT TB and iAdhere trials, which evaluated 3HP versus 9H in non-pregnant people, found that fetal loss and congenital anomalies were comparable among participants exposed to 3HP (n = 31) and 9H (n = 56) [58]. Moreover, the adverse events were similar to general rates in the United States. Taken together, these findings support that 3HP does not require a dose adjustment and is tolerable in pregnant people without HIV or those with HIV on efavirenz-based ART. Larger studies to assess 3HP safety in pregnant people on dolutegravir-based ART are planned. 

There are no data on 1HP in pregnant people. The BRIEF TB trial demonstrated 1HP is non-inferior to 9H in people with HIV, with a similar safety profile and higher level of adherence [59]. Pregnant women were excluded from this study, and no women became pregnant while taking 1HP. In the 9H arm, however, 136 women became pregnant, and there was an increase in composite adverse pregnancy outcomes in participants exposed to 9H in the first trimester [60]. 

In most trials of rifapentine-based TB preventive therapy in people with HIV, participants were on efavirenz-based regimens [59,61]. Currently, dolutegravir-based regimens are first-line, including for people who are pregnant and planning to conceive [62]. There are legitimate concerns about using rifapentine with dolutegravir because rifapentine induces enzymes which can decrease dolutegravir concentrations [63]. Moreover, pregnancy alone is associated with decreased dolutegravir concentrations [64]. So far, dose adjustments are not required for non-pregnant people on dolutegravir with 3HP nor for pregnant people on dolutegravir alone [63,64]. However, ACTG study A5372 found that doubling dolutegravir to twice-daily allowed adequate dolutegravir levels in people with HIV taking 1HP [65]. There are no data for pregnant people on dolutegravir and 3HP or 1HP. 

DOLPHIN-Moms is a prospective randomized trial that will study the safety of 1HP and 3HP taken with dolutegravir in pregnant people with HIV [66]. Moreover, it will also determine if twice-daily dolutegravir is required to maintain adequate levels when administered with 1HP or 3HP during pregnancy. If shown to be safe, the shorter duration of rifapentine-containing regimens could make them ideal for use in pregnancy because they can be completed during the antenatal period, when regular interaction with the healthcare system is common, and potentially avoid postpartum hepatotoxicity. 

Studies evaluating issues of TBI treatment in pregnant people primarily focus on isoniazid and are summarized in Table 3.

## 7. Timing of TBI Treatment in Pregnancy

The indication and timing of TBI treatment differ in high- and low-burden settings depending on HIV status [4,51,74]. Generally, in low-burden settings, TBI treatment is targeted to pregnant people with a positive IGRA or TST. US guidelines note that TBI treatment can be delayed until 2–3 months post-partum for people at lower risk of TB progression. If a person has had recent exposure to someone with infectious TB (or recent TBI test conversion), however, TBI treatment should be initiated immediately, even during the first trimester of pregnancy [51,74]. For pregnant people with HIV, US guidelines also recommend treatment deferral until after delivery if there are no close contacts with infectious TB [74]. In contrast, the WHO recommends that TB preventive therapy be given immediately to pregnant people with HIV living in high-burden settings (irrespective of recent contact), acknowledging that systematic postpartum deferral misses the point when they are most vulnerable to TB [4]. 

Key considerations for the timing of TBI treatment in pregnancy include immunologic changes of pregnancy that may affect TB susceptibility, potential teratogenicity of medications during early fetal development, physiologic changes that affect the pharmacokinetics, and the risk of hepatotoxicity in late pregnancy and early postpartum.

## 8. MDR Prevention in Pregnant People

People exposed to drug-resistant TB (DR-TB) have a high risk of developing DR-TB. A meta-analysis reported 47% of DR-TB household contacts developed TBI [75]. The risk is higher in young children and may be higher during pregnancy [76]. Prevention of DR-TB is especially important in pregnant people because there are no internationally accepted regimens for DR-TB treatment in pregnancy [77]. 

Currently, the WHO and US CDC recommend Levofloxacin for 6–12 months in non-pregnant DR-TB household contacts. There are no recommendations for DR-TB prevention during pregnancy nor are there currently any planned DR-TB prevention trials that include pregnant people. 

## 9. Research Priorities 

### Biomarkers of Progression from TB Infection to Disease

A biomarker for TB progression in pregnant or non-pregnant populations remains elusive. Multiple studies have been conducted to identify a transcriptional profile that accurately predicts TB progression in non-pregnant people but have all excluded pregnant people. Because of the immune changes of pregnancy, biomarkers for TB progression may be different than those in non-pregnant populations. A small transcriptional study in India identified a gene set associated with glutathione metabolism that predicted TB progression in pregnant women; it remains unknown if the signature will be validated in other pregnant cohorts. In Kenya and Uganda, nonspecific T-cell activation, a biomarker for TB disease development [78], increased from pregnancy to postpartum in women with TBI compared to without TB; this change did not necessarily predict TB disease [40].

Taken together, these data suggest immune changes of pregnancy may impair robust immune responses to Mtb, impacting progression from TB infection to disease through novel immune pathways. More definitive immunologic studies are needed. 

## 10. Conclusions 

Incorporating pregnancy and postpartum status into routine surveillance data can improve our understanding of TBI prevalence, TB risk, and outcomes. Other major gaps include whether pregnancy-related immunologic changes affect susceptibility of progression to TB disease and the identification of an immune correlate of TB risk. Optimal timing of TBI treatment must take into account the risk of TB progression as well as safety considerations unique to pregnancy. Identifying the safest regimens for TBI treatment for both drug-sensitive and drug-resistant TB, including for people with HIV, requires inclusion of pregnant people in TB prevention trials.

## Figures and Tables

**Table 2 pathogens-11-01481-t002:** Treatment regimens for latent TB infection in pregnancy.

Regimen	Dosing	Comment
**6H or 9H** Isoniazid * daily for 6 or 9 months	Isoniazid 5 mg/kg (300 mg maximum)	CDC/WHO: preferred for pregnant PWHIV
**36H** Isoniazid * daily for 36 months	Isoniazid 5 mg/kg (300 mg maximum)	WHO: preferred in settings of high TB transmission including pregnant PWHIV
**3HR** Isoniazid * AND Rifampin daily for 3 months	Isoniazid 5 mg/kg (300 mg maximum) AND Rifampin 10 mg/kg (600 mg maximum)	
**4R** Rifampin daily for 4 months	Rifampin 10 mg/kg (600 mg maximum)	
**3HP**^¶^ Isoniazid * AND Rifapentine weekly for 3 months	***Not currently recommended in pregnancy*** (some pregnancy safety data)	
**1HP**^¶^ Isoniazid * AND Rifapentine daily for 1 month	***Not currently recommended in pregnancy*** (no data in pregnancy, trials planned)	

Abbreviations: CDC = US Centers for Disease Control and Prevention; WHO = World Health Organization; PWHIV = people with HIV; **^*^** Isoniazid containing regimens should be administered with pyridoxine (B6) 25 to 50 mg daily to reduce risk of peripheral neuropathy; ^¶^ Data on safety of rifapentine in pregnancy are limited; therefore, 3HP and 1HP are not currently recommended in people who are pregnant or expecting to become pregnant during the treatment period.

**Table 3 pathogens-11-01481-t003:** Studies of latent TB infection treatment in pregnancy.

Study	Treatment Regimen	Study Site	Population	Study Design	Findings
Franks et al., Public Health Reports 1989 [67]	6H	USA	3681 women during and after pregnancy	Retrospective cohort	A total of 5 pregnant women developed hepatitis; 2 of the 5 died.
Moulding et al., Am Rev Resp Disease 1989 [68]	6H	USA	24 people who died while taking isoniazid	Retrospective cohort	A total of 20 isoniazid associated deaths, 4 initiated isoniazid in pregnancy.
Martinson et al., NEJM 2011 [69]	3HP vs. 3HR vs. 6H	South Africa	235 WHIV who became pregnant during treatment or follow-up	Sub-analysis of RCT	Pregnant women on 3HP and 3HR were discontinued off treatment. A total of 26 became pregnant on isoniazid; 10 chose to continue with no toxicity observed.
Taylor et al., IDOBGYN 2013 [54]	6H vs. 36H	Botswana	103 WHIV who became pregnant during trial (37% on cART, 63% on AZT or AZT/3TC for PMTCT)	Sub-analysis of RCT	No isoniazid-associated hepatitis or other severe isoniazid-associated adverse events were observed.
Tiam et al., JAIDS 2014 [70]	6H	Lesotho	160 Pregnant WHIV at 1st ANC visit (36% on ART, 65% on AZT for PMTCT)	Prospective cohort	IPT was initiated in 124/158 (78.5%) pregnant WHIV; 64.5% women completed a 6-month IPT regimen; 2 (1.6%) died of causes unrelated to IPT/TB; and 31.5% were lost to follow-up.
Moro et al., Ann Am Thorac Soc 2018 [58]	3HP vs. 9H	USA, Canada, Brazil, Spain, Peru, South Africa, Hong Kong	126 women who became pregnant during trial	Sub-analysis of 2 RCTs	Of the total 126 pregnancies, fetal loss was reported in 8/54 (15%) and 9/72 (13%), 3HP and 9H, respectively; and congenital anomalies in 1/37 (3%) and 2/56 (4%) live births, 3HP and 9H, respectively. The overall proportions of fetal loss (17/126 (13%)) and anomalies (3/93 (3%)) were similar to those estimated for the United States, 17% and 3%, respectively.
Gupta et al., NEJM 2019 [52]	6H	Botswana, Haiti, India, South Africa, Tanzania, Thailand, Uganda, Zimbabwe	956 pregnant WHIV, 14–34 weeks gestation, 99% on cART	RCT	A primary outcome event (treatment-related maternal adverse events of grade 3 or higher or permanent discontinuation of the trial regimen because of toxic effects) occurred in 72 of 477 women (15.1%) in the group that IPT was initiated in during pregnancy and in 73 of 479 (15.2%) in the group that IPT was initiated in postpartum. The risks associated with initiation of IPT during pregnancy appeared to be greater than those associated with initiation of therapy during the postpartum period.
LaCourse et al., 2019 [71]	6H	Kenya	300 postpartum WHIV, 100% on cART	Retrospective cohort	A total of 224 reported previous IPT; 155 (69%) had any IPT use during pregnancy. Six-month IPT completion rates were high (147/160 (91.9%)) among women with sufficient time to complete before infant trial enrollment and similar among preconception or during pregnancy initiators.
Kalk et al., Clin Infect Dis 2020 [53]	6H-12H	South Africa	43,971 Pregnant WHIV on or initiating cART	Retrospective cohort	A total of 16.6% received IPT during pregnancy. Women who received IPT were less likely to experience poor pregnancy outcomes (adjusted odds ratio (aOR), 0.83); this association strengthened with IPT started after the first trimester compared with none (aOR, 0.71) or with first-trimester exposure (aOR, 0.64). IPT reduced the risk of TB by approximately 30%.
Salazar-Austin et al., Clin Infect Dis 2020 [11]	6H	South Africa	151 Pregnant WHIV	Prospective cohort	Of the 69 IPT-exposed women, 11 (16%) had an adverse pregnancy outcome compared with 23 (28%) IPT-unexposed women. The adjusted odds of having an adverse pregnancy outcome were 2.5 times higher in IPT-unexposed women compared with IPT-exposed women after controlling for confounding factors.
Mathad et al., Clin Infect Dis 2022 [72]	3HP	Haiti, Kenya, Malawi, Thailand, Zimbabwe	50 pregnant w/ and w/o HIV	RCT	Among 30 women without HIV, clearance of rifapentine was 28% lower during pregnancy than postpartum. In 20 pregnant WHIV, clearance was 30% higher than women without HIV (*p* < 0.001). 3HP does not require dose adjustment in pregnancy. There were no drug-related serious adverse events.
Singh et al., JAIDS 2022 [73]	3HP vs. 3HR vs. 6H	South Africa	216 women who became pregnant during trial	Sub-analysis of RCT	A total of 216/896 women (24%) conceived during the study. A total of 34 women became pregnant while taking preventive treatment (8 rifamycin, 26 isoniazid monotherapy). The odds of pregnancy were higher in women in the rifamycin-isoniazid arms than the isoniazid arms (3HP: 1.73, *p* = 0.001; 3HR: 1.55, *p* = 0.017) despite increased contraceptive use compared to the standard 6H therapy.

Treatments: 6H–12H = Isoniazid daily for 6–12 months, 3HP = Isoniazid and Rifampin weekly for 3 months, 3HR = Isoniazid and Rifampin daily for 3 months, 36H = Isoniazid daily for 36 months. Abbreviations: WHIV = women with HIV; cART = combined antiretroviral therapy; ANC = antenatal care; IPT = Isoniazid preventative therapy; AZT = Azidothymidine; 3CT = Lamivudine; PMTCT = prevention of mother to child transmission; RCT = randomized controlled trial.

## Data Availability

Not applicable.

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
