# Peer review of "Tuberculosis Infection in Pregnant People: Current Practices and Research Priorities"

_pathogens, 2022, doi:10.3390/pathogens11121481_

Round 1
Reviewer 1 Report
Mathad and colleagues present a systematic review of the current situation of Tuberculosis (TB) incidence and treatments in pregnant people. They clearly summarize the global prevalence of the disease in high and low burden areas, and the problems posed by HIV coinfections. The authors also identify gaps in current knowledge. My comments are summarized below.
1. The authors talk about TB infection, which I presume is the pulmonary form of the disease. Please mention this in the manuscript. It is not clear whether they also include extra pulmonary TB in their discussions.
2. I suggest rearranging the sections. As written originally, the Introduction (1), Epidemiology (2), and Natural history (3) looks good. These can be followed by Diagnostics (5), infection treatment (6), newer short course (7), and timing of treatment (9). MDR prevention (8) can then follow. The manuscript can then end with the Research Priorities such as biomarkers (4) and a final conclusion section summarizing the authors’ vision for future research in this area.
3. Line 46: The authors discuss outcomes of maternal TB disease can be exacerbated by HIV, which is the major comorbidity they address in the manuscript. Can the authors add few sentences about other situations that may contribute to poor outcomes associated with maternal TB such as malnutrition, lack of access to healthcare (especially during the COVID pandemic), and other comorbidities.
Minor comments:
1. Line 71: add immune response dampening cells
2. Please be consistent with Mtb abbreviation. Table 1, page 4 has M. TB instead of Mtb mentioned in the text.
3. Line 73: Change progress to progresses
4. Line 102: Add Current tests to detect TBI
5. Line 117: Correction IGRA+/TST(?)
6. Line 211: correction to ‘pregnancy outcomes when’
7. Line 255: Typo ‘For’.
Author Response
- The authors talk about TB infection, which I presume is the pulmonary form of the disease. Please mention this in the manuscript. It is not clear whether they also include extra pulmonary TB in their discussions.
RESPONSE: (SYLVIA) The term TB infection refers to This invited manuscript is part of a Special Edition focused on multiple areas of TB infection. While an in depth discussion of TB infection versus disease is outside the scope of this specific manuscript, we have clarified that TB infection is a “(a state of persistent immune response to stimulation by Mycobacterium tuberculosis (Mtb) antigens without evidence of clinically active TB)” in the Introduction.
- I suggest rearranging the sections. As written originally, the Introduction (1), Epidemiology (2), and Natural history (3) looks good. These can be followed by Diagnostics (5), infection treatment (6), newer short course (7), and timing of treatment (9). MDR prevention (8) can then follow. The manuscript can then end with the Research Priorities such as biomarkers (4) and a final conclusion section summarizing the authors’ vision for future research in this area.
RESPONSE: (SYLVIA) We thank the Reviewer for this suggestion. Sections have been rearranged as suggested, including the addition of a final conclusions section.
- Line 46: The authors discuss outcomes of maternal TB disease can be exacerbated by HIV, which is the major comorbidity they address in the manuscript. Can the authors add few sentences about other situations that may contribute to poor outcomes associated with maternal TB such as malnutrition, lack of access to healthcare (especially during the COVID pandemic), and other comorbidities.
RESPONSE: (JYOTI)Thank you; We have added, “Similar to non-pregnant populations, lack of access or delay in accessing health care also contribute to poor outcomes. Malnutrition, alcohol or tobacco abuse, and diabetes may also contribute to poor TB outcomes during pregnancy, but there are limited data on these risk factors in pregnant people”.
Minor comments:
RESPONSE: (SYLVIA) Thank you for these suggested edits below which we have incorporated.
- Line 71: add immune response dampening cells
- Please be consistent with Mtb abbreviation. Table 1, page 4 has M. TB instead of Mtb mentioned in the text.
- Line 73: Change progress to progresses
- Line 102: Add Current tests to detect TBI
- Line 117: Correction IGRA+/TST(?)
- Line 211: correction to ‘pregnancy outcomes when’
- Line 255: Typo ‘For’.
Reviewer 2 Report
In this review, the authors have discussed about the global incidence of TB in pregnant women. Tuberculosis is a global threat causing 2 million deaths every year. About one third of the human population are carrier of this disease. Although a lot of work have been done in characterization and treatment of Mycobacterium tuberculosis, not much work has been conducted in the field of TB in pregnant women. This review addresses this important issue about the association of TB with pregnancy.
The authors have also emphasized about the lack of Biosensors to detect during pregnancy. This is a very important issue that needs to be addressed as development of TB during pregnancy can have fatal consequences including transmission of TB to the new-born as well as giving birth to an immunocompromised child. Further, the authors have discussed about the consequences of TB infection of a pregnant women infected with HIV. There needs to be a lot of work in this field as the HIV infected women are unable to elicit an IFN-g immune response to the pathogen.
Furthermore, the authors have also discussed about the different treatment regimens of antibiotics that can be consumed by the pregnant women to treat tuberculosis. Of special interest is lack of data for the newer short course of TB treatment involving use of isoniazid and rifapentine. Further work needs to be done for safe administration of antibiotics to treat tuberculosis in pregnant women.
Overall, the authors have highlighted a very important but less studied area of tuberculosis research. There is an ardent need for doing more research and generating data for antibiotic treatment of pregnant women who are infected with TB. The review is well written, informative and should be of interest to readers studying tuberculosis.
I’ve a minor comment that the authors should address before acceptance:
1. The authors should allocate a section about the challenges associated with treatment of pregnant women who have diabetes when infected with Tuberculosis. Several labs have published data on association of diabetes and its compounding effect when a person get infected with tuberculosis. This is a new area of research and incorporation should enrich the review.
2. The authors should also try to incorporate an image about the immune response of pregnant women infected with TB verses a healthy woman as it will be easier to grasp those underlying challenges of TB treatment during pregnancy.
Author Response
- The authors should allocate a section about the challenges associated with treatment of pregnant women who have diabetes when infected with Tuberculosis. Several labs have published data on association of diabetes and its compounding effect when a person get infected with tuberculosis. This is a new area of research and incorporation should enrich the review.
RESPONSE: (JYOTI)We appreciate this comment and agree that it is an important issue. However, most studies of diabetes and tuberculosis exclude pregnant people. We have added a short sentence about one small study that suggests that gestational diabetes could increase the risk of TB progression. (lines 119-124)
- The authors should also try to incorporate an image about the immune response of pregnant women infected with TB verses a healthy woman as it will be easier to grasp those underlying challenges of TB treatment during pregnancy.
RESPONSE: (Sylvia) Thank you for this excellent suggestion. Unfortunately, there are many gaps in our knowledge regarding the immune responses of pregnancy and their role in TB pathogenesis. We have referenced an excellent review article that outlines general immunologic changes in pregnancy (Kourtis et al. NEJM 2014) as well as provided a brief overview of what changes may be associated with TBI susceptibility throughout our manuscript . A comprehensive figure is beyond the scope of our review.